# Serotonergic neuronal death and concomitant serotonin deficiency curb copulation ability of *Drosophila platonic* mutants

Yasemin B. Yilmazer[1], Masayuki Koganezawa[1], Kosei Sato[1], Jinhua Xu[2] & Daisuke Yamamoto[1]

*Drosophila platonic* (*plt*) males court females, but fail to copulate. Here we show that *plt* is an allele of *scribbler* (*sbb*), a BMP signalling component. *sbb* knockdown in larvae leads to the loss of approximately eight serotonergic neurons, which express the sex-determinant protein Doublesex (Dsx). Genetic deprivation of serotonin (5-HT) from *dsx*-expressing neurons results in copulation defects. Thus, $sbb^+$ is developmentally required for the survival of a specific subset of *dsx*-expressing neurons, which support the normal execution of copulation in adults by providing 5-HT. Our study highlights the conserved involvement of serotonergic neurons in the control of copulatory mechanisms and the key role of BMP signalling in the formation of a sex-specific circuitry.

[1] Division of Neurogenetics, Tohoku University Graduate School of Life Sciences, 2-1-1 Katahira, Aoba-ku, Sendai 980-8577, Japan. [2] School of Medicine and Key Laboratory of Optoelectronic Chemical Materials and Devices, Ministry of Education, Jianghan University, Wuhan, Hubei 430056, China. Correspondence and requests for materials should be addressed to D.Y. (email: daichan@m.tohoku.ac.jp).

When successful, male courtship culminates in copulation, yet the neural mechanism underlying copulation has attracted limited attention in biology[1]. Indeed, copulation represents a complex behaviour that consists of coordinated motor acts involving the entire body, and thus requires comprehensive analyses across the molecular, cellular and organismal levels to clarify its neural basis[2]. One of the promising approaches to such investigation would be genetic dissections, which start with isolating mutants with specific defects in copulation. Indeed, mutants with defects of copulation have been identified. *coitus interruptus* (*coi*) mutant males have shorter copulation durations than wild-type males do[3]. Males of *stuck* (*sk*)[3] and *lingerer* (*lig*)[4] mutants often fail in withdrawing their genitalia to terminate copulation. *lig* mutant males also less frequently attempt copulation despite their normal levels of overall courtship activities[4]. Although these mutations unravelled some of mechanistic elements composing the copulation motor program, they provided fewer insights into the neural basis for copulation. Recent technical advances have allowed the artificial activation and/or inactivation of selected neurons *in vivo*, opening a route to determination of the circuitries underlying copulation behaviour. This approach led to the identification of pC2l, a *dsx*-expressing (*dsx*[+]) neural cluster in the brain, which plays a role in triggering attempted copulation[5], as well as eight GABAergic *dsx*[+] neurons in the abdominal ganglion with roles in regulating the copulation duration[6]. The GABAergic regulation of copulation duration is antagonized by the dopaminergic system in the abdominal ganglion[6]. In addition, insulin-producing cells (IPCs) in the pars intercerebralis, a neurosecretory centre of the insect brain, have been shown to regulate the timing and frequency of attempted copulation; serotonin receptor 1A ($5\text{-}HT_{1A}$) knockdown in IPCs reduces the latency to abdominal curling and increases the incidence of attempted copulation[7]. $5\text{-}HT_{1A}$ knockdown in IPCs also shortens the latency to courtship initiation[8], implying its role as a motivational regulator. Thus, copulation is controlled by a multilayered system that contains, at very least, a brain centre (that is, pC2l) for triggering copulation attempts, a motivational modulator in the brain (that is, IPCs), an abdominal centre for tuning copulation persistence, and an as yet unidentified central pattern generator for the execution of copulation, which is presumably located in the ventral nerve cord (the thoracic and abdominal ganglia[9]). Starting with the isolation of a mutant named *platonic* (*plt*), whose males fail to copulate despite vigorous courtship towards a female (reviewed in refs 2,10), here we identify a hitherto uncovered component of the copulation-controlling *dsx*[+] neural system, which functions to improve motor performance for copulation by secreting serotonin (5-HT) in the abdominal ganglion. The *plt* mutation impairs the BMP signalling required for the survival of eight serotonergic interneurons in the abdominal ganglion, and thereby induces copulation defects in mutant males via a 5-HT deficiency. Our results highlight the conserved roles of 5-HT in the regulation of copulation and of BMP signalling in the development of cellular substrates for sex-specific behaviours.

## Results

***plt* is an allele of *sbb***. *plt* homozygotes are semilethal, and exhibit a visible phenotype in the wing vein: the posterior-most wing vein fails to reach the wing margin (Supplementary Fig. 1). The *plt* mutant line carries an insertion of a $ry^+$-marked P-element at the cytological location 53C. Outcrosses over five generations to a $ry^-$ control line led to the segregation of neither the copulation defect nor the wing vein defect from the $ry^+$ marker, strongly

suggesting that the two apparently unrelated phenotypes are both linked to the same P-element insertion at 53C. Subsequent deficiency mapping verified this idea: two deficiencies, *Df(2R)BSC334* and *Df(2R)ED3683*, when placed *in trans* to *plt*, uncovered the no-copulation phenotype as well as the short-wing vein phenotype in these *plt* hemizogotes, whereas four other deficiencies, (*Df(2R)BSC335*, *Df(2R)BSC399*, *Df(2R)BSC483* and *Df(2R)Exel7153*), similarly tested failed to uncover either phenotype (Fig. 1a; Supplementary Fig. 1). The cytological breakpoints of these deficiencies defined a $\sim 70$-kb region within which the genetic sequences responsible for the two phenotypes likely exists (Fig. 1b). Searches of the *Drosophila* genome database revealed a candidate gene annotated to this $\sim 70$-kb region, *scribbler* (*sbb*). Indeed, genetic complementation tests where *plt* was combined with different *sbb* mutations demonstrated that *plt* is allelic to *sbb*: these heteroallelic mutants manifested both the copulation defect and wing vein defect (Fig. 1a; Supplementary Fig. 1). *plt* is thus referred to as $sbb^{plt}$ hereafter. Moreover, inverse PCR of the genomic sequence flanking the mutagenic P-element demonstrated that the P-element was inserted 31-bp downstream of the *sbb* transcription start site (Fig. 1c). As expected, $sbb^{plt}$ hemizygous mutants exhibited a reduction in *sbb* messenger RNA (mRNA) expression as detected by PCR with reverse transcription (Supplementary Fig. 2a). We conclude that the impairment in copulation and wing vein formation in $sbb^{plt}$ resulted from the reduction in *sbb* expression.

*sbb* knockdown in the entire body by the combination of *hs-GAL4* and *UAS-sbbRNAi* induced both the copulation defect and wing vein defect characteristic of $sbb^{plt}$ hemizygous mutants, provided that heat shock was applied at an appropriate time point (Fig. 1d; Supplementary Fig. 3). PCR with reverse transcription also confirmed that *UAS-sbbRNAi* can effectively reduce *sbb* mRNA levels (Supplementary Fig. 2b). Interestingly, the most effective time point for heat shock was 4 days after egg laying for both of the phenotypes (Fig. 1c; Supplementary Fig. 3), indicating that $sbb^+$ is required in the larval stage to achieve acquisition of a normal wing vein and normal copulation in the adult fly. This result is compatible with the known developmental roles of Sbb, which functions as a transcription factor in signal transduction downstream of Dpp, a conserved BMP family morphogen[11,12].

**Neural origin of copulation defects in *plt* mutants**. Because our anatomical analysis revealed no obvious structural anomalies in the external genitalia and associated muscles in $sbb^{plt}$ hemizygous mutants (Supplementary Fig. 4), the unsuccessful copulation in these males likely originated from neural malfunction. $sbb^{plt}$ hemizygous mutants gave high courtship indices (Fig. 1a; Supplementary Fig. 5a) displaying wing extension/vibration at high rates (Supplementary Fig. 5b), yet having a markedly reduced incidence of licking and attempted copulation (Supplementary Fig. 5c,d). These observations support the notion that the copulation defect has a neural origin.

Because the major part of the male courtship circuitry has been shown to be composed of neurons that express either or both of the two sex determination genes *fruitless* (*fru*) and *doublesex* (*dsx*)[13–15], we examined the effects of *sbb* knockdown with *UAS-sbbRNAi* as driven by *fru-GAL4* ($fru^{GAL4}$ or $fru^{NP21}$) or *dsx-GAL4* ($dsx^{GAL4}$). In contrast to the partial suppression of copulation success with *fru-GAL4* (Fig. 2a; Supplementary Fig. 6), *dsx-GAL4* completely blocked copulation when used to express *UAS-sbbRNAi* (Fig. 2b), with only moderate effects on courtship activities (Supplementary Fig. 7a,b). The male flies of all test genotypes actively courted females (Supplementary Fig. 7a). Because *dsx*—unlike neuron-specific *fru*—is expressed in a wide

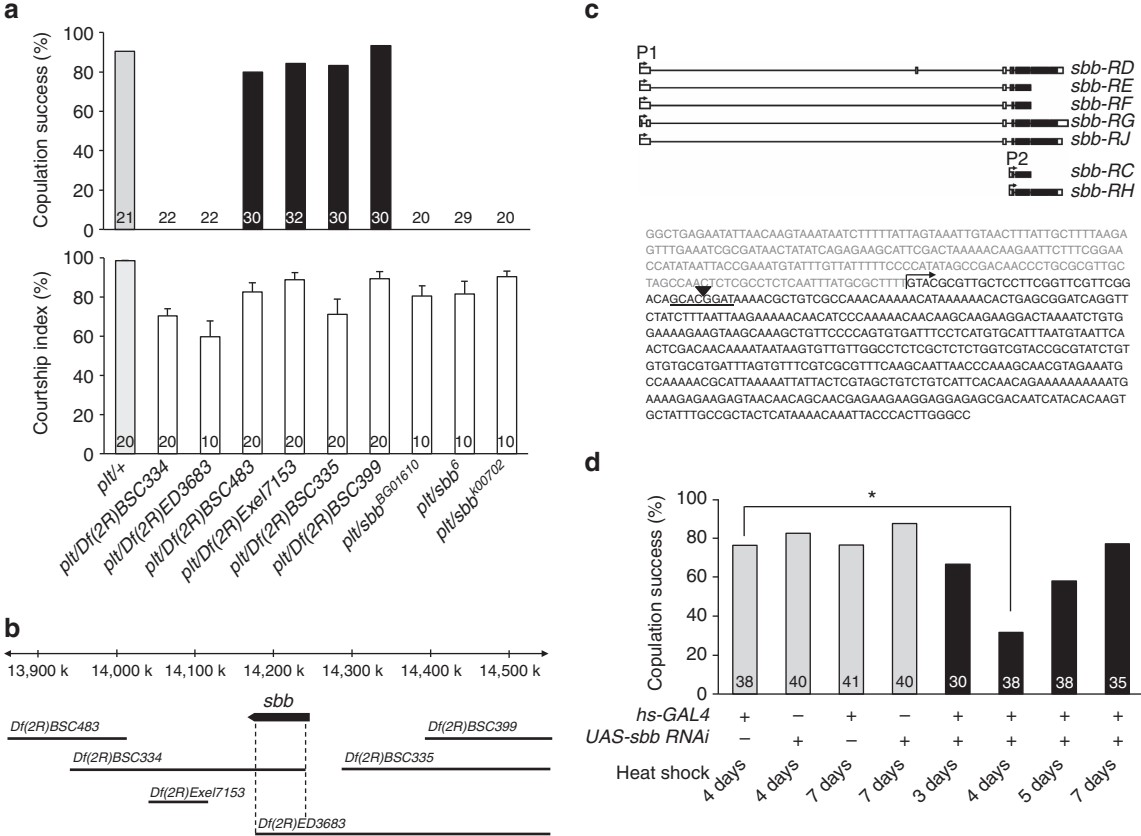

**Figure 1 | Mapping the *plt* mutation.** (**a**) Courtship activities (lower panel) and copulation success (upper panel) for the indicated mutant and control flies. In the lower panel of **a**, the mean ± s.e.m. values are shown. (**b**) Deficiency mapping of the *plt* mutation onto the genome. The top horizontal bar indicates the physical distance (in kb) relative to the *sbb* transcription start site. The *sbb*-transcribed region is indicated with an arrow, the direction of which indicates the direction of transcription. The deleted region for each deficiency chromosome is shown below the arrow (horizontal bars). (**c**) The genomic organization of the *sbb* gene (upper panel) and the sequence determined by inverse PCR (lower panel). In the upper panel, exons (black boxes: coding regions; open boxes: non-coding regions) and introns (thin horizontal lines) for several transcripts are shown. In the lower panel, the mutagenic *P*-element insertion site (an inverted triangle), the target sequence of the *P*-element (underlined) and the transcription start site are indicated. (**d**) Effect on the copulation rate of developmental timing of *sbb* knockdown. The transgenes carried by tested flies are indicated below each bar (+, with the transgene; −, without the transgene). In all graphs, the number of flies examined is indicated below each bar. In **d**, statistical differences are evaluated by the $\chi^2$-test (*$P < 0.05$).

variety of non-neural as well as neural tissues, we evaluated the relative contribution of non-neural and neural tissues to the *sbbRNAi*-induced copulation block. When the *dsx-GAL4* activity was antagonized by GAL80 expressed selectively in neurons with neuron-specific *elav-GAL80*, the copulation block by *sbbRNAi* was perfectly abrogated (Fig. 2c; Supplementary Fig. 8a), indicating that the no-copulation phenotype is ascribable to the *sbb* knockdown in *dsx*-expressing neurons but not non-neural *dsx*-positive tissues. To further ascertain whether neuronal *sbb+* is a requirement for normal copulation, *dsx*-positive glia were excluded from *sbbRNAi* expression with the aid of *repo-GAL80* (the *repo* promoter is specifically active in glia). This result indicated that *sbbRNAi* expression via *dsx-GAL4* completely blocked copulation irrespective of whether *repo-GAL80* was present or not (Fig. 2d; Supplementary Fig. 8e), indicating that glial cells are not involved in this phenotype.

Some neurons express both *dsx* and *fru* (*dsx/fru* double-positive cells: *dsx[+]/fru[+]*), while others express either *dsx* or *fru* (*dsx* single-positive cells: *dsx[+]/fru[−]*; *fru*-single-positive cells: *dsx[−]/fru[+]*), and the rest express neither of these genes[16–18]. To determine which type of *dsx*-positive cells requires *sbb+* for normal copulation, we restricted the action of *dsx-GAL4* by expressing GAL80 in a subset of *dsx*-positive cells using three

different intersection strategies. In the first strategy, we used a conditional GAL80 transgene, *tubP > GAL80 >*. Here the two > indicate the FRT sequences, that is, the target of the flippase FLP; the FLP binds to the FRTs flanking *GAL80*, thereby excising *GAL80* and allowing GAL4 to act. We combined *tub > GAL80 >* with *fru^FLP*, which expresses FLP from the endogenous *fru* gene promoter, and in this way activated GAL4 only within *dsx[+]/fru[+]* cells (Fig. 2e). In the second strategy, we used the other conditional *GAL80* transgene, *tubP > stop > GAL80*, in conjunction with *dsx^FLP*, which expresses FLP via the endogenous *dsx* gene promoter. Because GAL80 is expressed only when FLP has excised the stop sequence between the *tub* promoter and the *GAL80*-coding region, this transgene allowed us to restrict the activation of GAL4 to only *dsx[−]/fru[+]* cells when combined with *dsx^FLP* (Fig. 2f). In the third strategy, *lexAop-GAL80* was used in the presence of *fru^LexA*. In this case, the LexA transcription factor is produced only in cells where an endogenous *fru* gene promoter is active, and the resulting LexA binds to *lexAop*, leading to the generation of GAL80 in these cells, and thus to GAL4 activation only in *dsx[+]/fru[−]* cells (Fig. 2g). In these three types of experiment, the GAL4 activity was restricted to the cell group as planned, judging from the reporter expression (Supplementary Fig. 8b–d). We found that

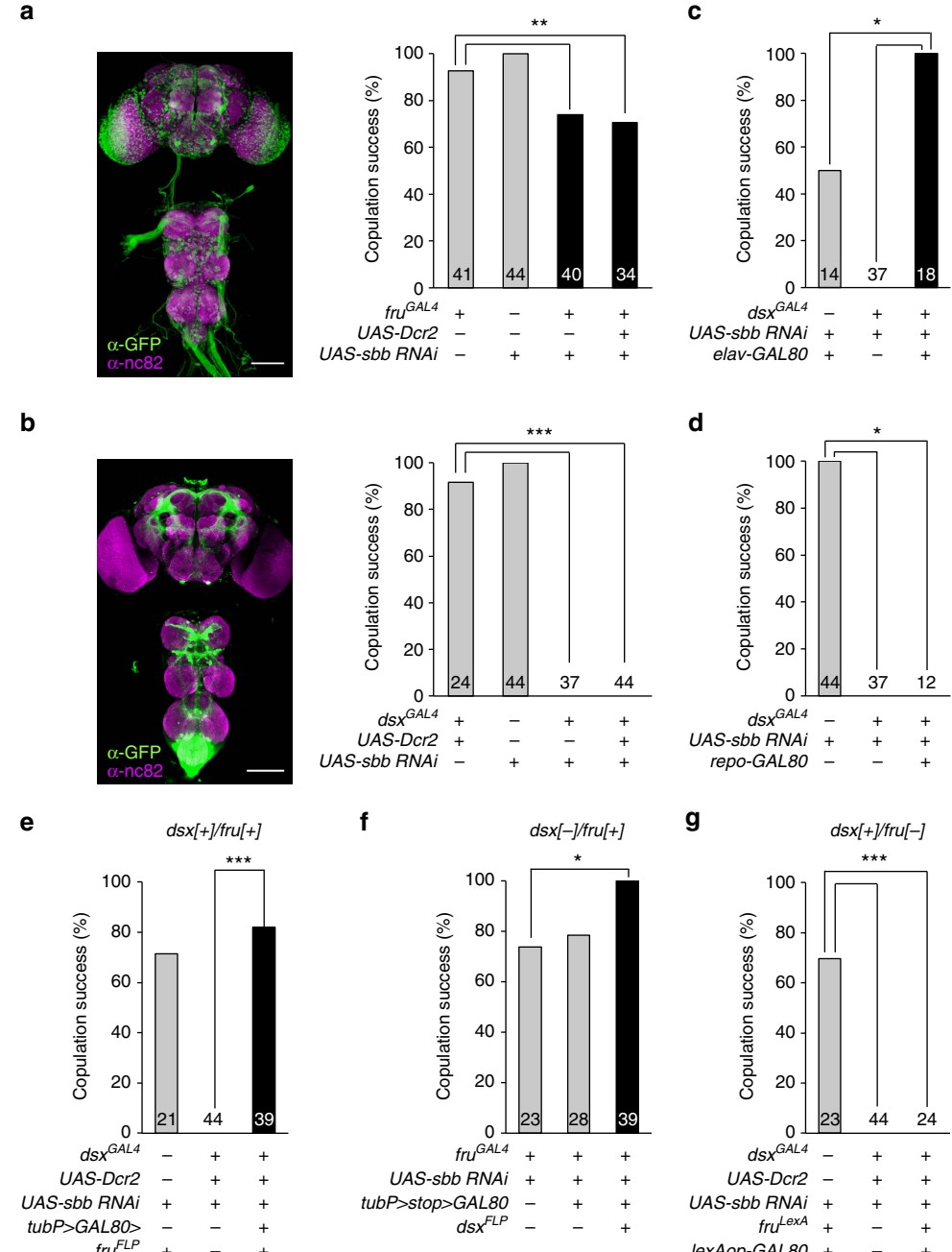

**Figure 2 | Contribution of neurons positive for *fru* and/or *dsx* and of glia to the copulation defect in flies with *sbb* knockdown.** Images (left-side panels) of the CNS with *fru*-positive (**a**) or *dsx*-positive (**b**) neurons and the effect of *sbb* knockdown in the respective neurons (right-side panels) on the copulation rate. The genotypes of flies for the histological analysis are *w; UAS-mCD8::GFP/+ ; fru^{GAL4}/+* (**a**) and *w; UAS-mCD8::GFP/+ ; dsx^{GAL4}/+* (**b**), respectively. Scale bar, 100 µm. (**c,d**) Effect of neuron-excluded (**c**) or glia-excluded (**d**) *sbb* knockdown on copulation. (**e–g**) Effect of *sbb* knockdown in *dsx*[ + ]/*fru*[ + ] (**e**), *dsx*[ − ]/*fru*[ + ] (**f**) or *dsx*[ + ]/*fru*[ − ] (**g**) on copulation. *P < 0.05; **P < 0.01; ***P < 0.001 by the $\chi^2$-test. Data are presented as described in the legend of Fig. 1.

the no-copulation phenotype was attained only when *sbbRNAi* was expressed in *dsx*[ + ]/*fru*[ − ] cells (Fig. 2e–g), indicating that *dsx* single-positive neurons are critically involved in this phenotype.

**sbb is required in abdominal *dsx* neurons for normal copulation.** To further narrow down the neuron groups in which *sbb^+* is required for normal copulation, we restricted GAL4 activity to different fractions of *dsx*-positive cells using *Otd-FLP*, which

expresses FLP only in the head, using *tsh-GAL80*, which prohibits GAL4 activity in the thorax, or selecting *GAL4*s that preferentially drive expression in particular groups of neurons that are identified by neurotransmitters. We also tested *GAL4* drivers for chemo-sensory cells (*poxn-GAL4*) or mechanosensory cells (*ppk-GAL4*), as these peripheral neurons may guide a male fly (or the male genitalia) to potential copulation targets[19]. When *sbb* knockdown was targeted to all the central nervous system (CNS) *dsx* cells except those in the head, the copulation defect was still produced (Fig. 3a; Supplementary Fig. 8f), whereas head-specific *sbb*

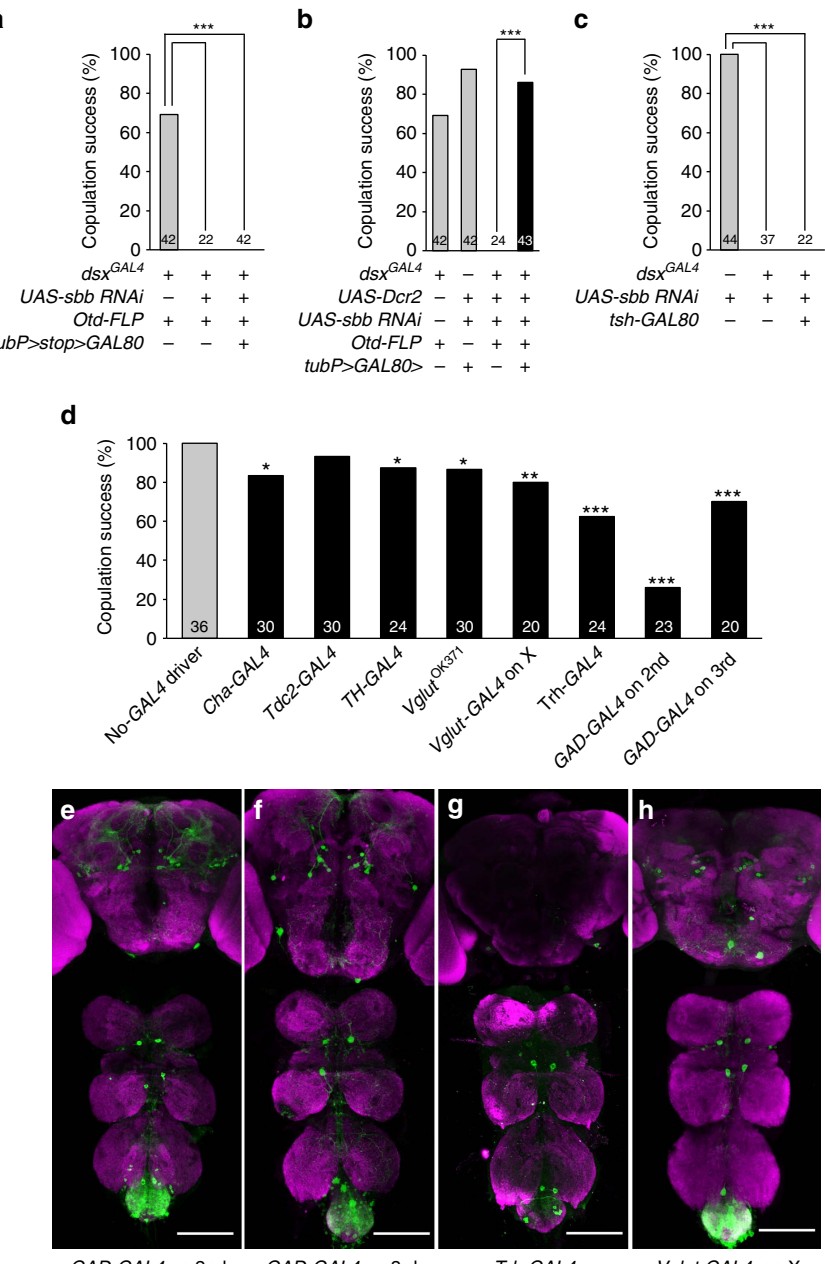

**Figure 3 | Effect of *sbb* knockdown in subsets of *dsx* neurons as defined by their body region or by the transmitters they synthesize.** Effect of head-excluded (**a**), head-specific (**b**) and thorax-excluded *sbb* knockdown in *dsx*-expressing neurons (**c**) on copulation. (**d**) Comparisons of *GAL4*s defined by transmitters in terms of their effect on copulation when used to drive *UAS-sbbRNAi* only in *dsx*-expressing cells by means of *tubP>GAL80>* and *dsx^FLP^*. (**e–h**) Images of *dsx* neurons that express *GAD-GAL4* on 2nd (**e**), *GAD-GAL4* on 3rd (**f**), or *Trh-GAL4* (**g**) or *Vglut-GAL4* on X (**h**). The genotypes of flies were *w; GAD-GAL4 on 2nd/UAS>stop>mCD8::GFP; dsx^FLP^/+* (**e**), *w; UAS>stop>mCD8::GFP/+; dsx^FLP^/GAD-GAL4 on 3rd* (**f**), *w; Trh-GAL4/ UAS>stop>mCD8::GFP; dsx^FLP^/+* (**g**) and *Vglut-GAL4 on X; UAS>stop>mCD8::GFP/+; dsx^FLP^/+* (**h**), respectively. *P = 0.05; **P = 0.01; ***P = 0.001 Scale bar, 100 μm.

knockdown did not impair copulation (Fig. 3b; Supplementary Fig. 8g); these results indicate that the *dsx*-expressing cells in the brain are not one of the sites at which *sbb*⁺ may be required for normal copulation. *sbb* knockdown in *dsx*-expressing cells in the CNS excluding those in the thorax still induced the copulation defect (Fig. 3c; Supplementary Fig. 8h); this result indicates that the thoracic *dsx*-expressing cells are not a candidate site for inducing this phenotype. Note, however, that the thoracic *dsx* neuron cluster called TN2 escaped the action of *tsh-GAL80* for an unknown reason, and thus this group of *dsx*-expressing neurons is not excluded from the candidate sites at which *sbb*⁺ may function for

copulation. In addition, a pair of large *dsx* neurons in the suboesophageal ganglion (SN) was not amenable to testing due to the lack of tools for its specific manipulation. These results collectively indicate that the *dsx*-expressing neurons in the non-thoracic ventral nerve cord (VNC), *dsx*-positive TN2 and *dsx*-positive SN remain as candidate cells. Peripheral mechano- and chemo-sensory neurons were judged not to make a large contribution to the no-copulation phenotype induced by *sbb* knockdown (Supplementary Fig. 9). Eight neurotransmitter-defined *GAL4*s representing six neurotransmitter species were tested in conjunction with *dsx^FLP^* and *tubP>GAL80>*:

acetylcholine (*Cha-GAL4*), octopamine (*Tdc2-GAL4*), dopamine (*TH-GAL4*), 5-HT (*Trh-GAL4*), L-glutamate (*Vglut[0k371]-GAL4* and *Vglut-GAL4* on X) and GABA (γ-aminobutyric acid; *GAD-GAL4* on 2nd and *GAD-GAL4* on 3rd). The most striking inhibition of copulation was observed when *sbbRNAi* was expressed under the control of *Trh-GAL4* or *GAD-GAL4*, although *Cha-GAL4*, *TH-GAL4* and *Vglut-GAL4* also yielded significant reductions in copulation success when used as drivers for *sbbRNAi* expression (Fig. 3d; Supplementary Figs 7c, 10 and 11). For further functional analysis, we used *Trh-GAL4*, which drove expression in a relatively small number of VNC cells compared with the other seven *GAL4*s, each of which yielded expression in hundreds of VNC cells, yet *Trh-GAL4* was the most effective of the *GAL4* drivers of *sbbRNAi* expression tested in terms of inhibiting copulation. We assumed that a small number of *Trh-GAL4*-positive neurons may have a decisive effect on copulation success. Although *Trh-GAL4* was used with the intention of labelling and manipulating serotonergic cells, the putative *cis*-region of a gene fused with *GAL4* does not necessarily recapitulate the endogenous expression of the gene in transformant flies. In fact, two *GAD-GAL4* lines used in this study revealed noticeably different expression patterns of a reporter in the CNS (Fig. 3e,f). We therefore performed double staining of the CNS for the expression of the *Trh-GAL4*-driven reporter and the immunoreactivity to an anti-5-HT antibody, and found that the *Trh-GAL4* expression coincides, in principle, with the anti-5-HT immunoreactivity (Supplementary Fig. 10c). However, there were also discrepancies between the *Trh-GAL4* expression and localization of anti-5-HT immunoreactivity. For example, '*tsh-GAL80*-resistant' TN2 were positive for *Trh-GAL4* but negative for 5-HT (Supplementary Fig. 10c). In addition, TN2 and '*Otd-FLP*-insensitive' SN were labelled by two other *GAL4*s, *GAD-GAL4* and *Vglut-GAL4* (Fig. 3e,h; Supplementary Fig. 10a,b), although both TN2 and SN were negative for anti-GABA immunoreactivity (Supplementary Fig. 11a,b). We were unsuccessful in staining the VNC with antibodies against L-glutamate or Vglut. Next, therefore, we attempted to elucidate a key role of the serotonergic neurons in successful copulation using an experiment to examine the effect of deprivation and supplementation of 5-HT on copulation (see below). It is noteworthy that only a total of eight *dsx*-expressing cells were labelled by the intersection of *Trh-GAL4* and *dsx[FLP]* (Fig. 3g), even though *sbbRNAi* expression in these eight cells resulted in a marked reduction in copulation success (Fig. 3d). We therefore considered that the *Trh-GAL4*-positive *dsx* neurons may be promising candidates for the site of *sbb[+]* action in the control of copulation, although we cannot completely exclude the possibility that *sbb[+]* activity is also required in other cells, including SN and TN2, for normal copulation.

**Serotonergic neuronal death results in *plt* copulation defects.** An obvious question here is why the partial loss of function in *sbb* at the larval stage impairs copulation performance in the adult stage. Immunostaining of *Trh-GAL4*-positive *dsx*-expressing neurons in flies that also carried *UAS-sbbRNAi* revealed that nearly half of these cells were lost by *sbb* knockdown ($3.6 \pm 1.2$, $n = 5$ in *sbb* knockdown versus $7.7 \pm 0.7$, $n = 9$ in the control; Fig. 4a,b). Different sets of cells seemed to be lost in different flies with *sbb* knockdown. Thus, it is plausible that the presence of eight serotonergic *dsx*-positive neurons in the abdominal ganglion is essential for normal copulation to occur. Indeed, double staining for *Trh-GAL4*-driven mCD8::GFP and 5-HT immunoreactivity in the presence of *dsx[FLP]* highlighted these eight *Trh*-positive *dsx* neurons (Fig. 4c; Supplementary Fig. 10c), indicating that these cells express 5-HT. We consider that *sbb[+]* is required in the serotonergic *dsx*-positive cells for their normal development. Failures in this process likely trigger degeneration of some

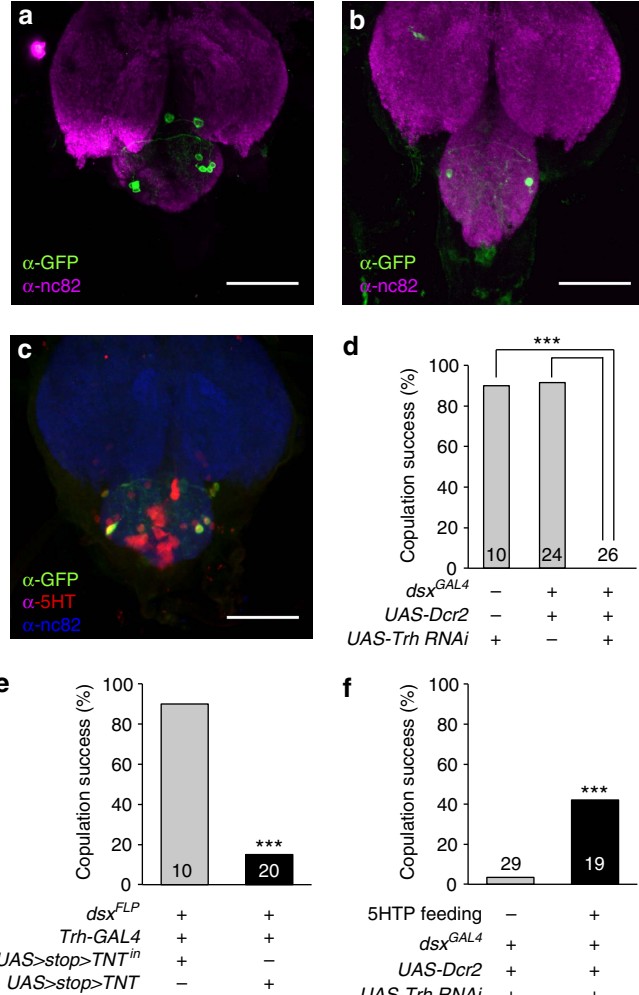

**Figure 4 | Roles of serotonergic *dsx* neurons and 5-HT in copulation.** Images of *Trh-GAL4*-positive *dsx* neurons in the abdominal ganglion in male flies without (**a**) and with (**b**) *sbb* knockdown. The genotypes of flies were *w; Trh-GAL4/UAS > stop > mCD8::GFP; dsx[FLP]/+* (**a**) and *w; Trh-GAL4/UAS-sbbRNAi tubP > GAL80 >; dsx[FLP]/UAS > stop > mCD8::GFP* (**b**), respectively. Scale bar, 50 μm. (**c**) *Trh-GAL4*-positive *dsx* neurons are immunopositive for an anti-5-HT antibody. Scale bar, 50 μm. (**d**) *Trh* knockdown in *dsx*-positive neurons completely blocks copulation. (**e**) Block of synaptic outputs from serotonergic *dsx* neurons by *TNT* inhibits copulation. Inactive *TNT* (*TNT[in]*) has no effect. (**f**) Copulation defects are rescued by feeding adults on 5-HTP-supplemented foods. 5-HTP was administered at 100 mM according to ref. 20. The mean ± s.e.m. and the number of flies examined are shown. ***$P < 0.001$ by the $\chi^2$-test.

of these cells with a consequence of 5-HT deficiency. Careful fluorescent microscopic examinations of these eight neurons as labelled by mCD8::GFP did not reveal peripheral axons, indicating that they are all interneurons.

To evaluate the importance of 5-HT in *dsx*-expressing cells in male copulation performance, we knocked *Trh* down in these cells, which exhibited a reduction in *trh* mRNA expression and a decrease in the number of cells with detectable 5-HT immunoreactivity (Supplementary Fig. 12). Intriguingly, *Trh* knockdown in all *dsx*-expressing cells completely blocked copulation (Fig. 4d), reminiscent of the *sbb[plt]* mutant phenotype. Block of synaptic outputs from these neurons with Tetanus toxin light chain (TNT) also inhibited copulation (Fig. 4e; Supplementary Fig. 7e). In contrast, activation via dTrpA1 of the same set of neurons was without effect: copulation success in the presence of a female

was not increased (75% at 19 °C, 71% at 32 °C; $n = 24$) nor did a solitary male display motions associated with copulation. This might mean that 5-HT secreted by these neurons gates execution of copulation when the released dose exceeds a threshold, but elicits no further effect at higher doses beyond the threshold. To determine whether 5-HT is necessary for executing copulation at the adult stage or for normal development of neural components, we examined the effect of supplementing foods with the 5-HT precursor 5-hydroxytryptophan (5-HTP) after adult emergence[20]. When flies with *Trh* knockdown in *dsx*-expressing cells were fed with 5-HTP only at the adult stage, the emerged adult males exhibited higher copulation success (Fig. 4f; Supplementary Fig. 7d), accompanied by an increase in the number of cells with detectable 5-HT immunoreactivity in *dsx-GAL4*-positive cells (Supplementary Fig. 12). These results demonstrate that 5-HT functions in the control of copulation at the adult stage.

## Discussion

Overall, our results indicate that *sbb*[+] at the larval stage supports the survival of a subset of *dsx*-positive VNC neurons that produce 5-HT, which is essential for copulation to occur at the adult stage. The *sbb* gene encodes C2H2-type zinc-finger proteins that function downstream of the major morphogen Dpp[11,12], and thus plays roles in a variety of cellular events, including those pivotal for neural developments such as axon guidance[21]. In this paper, we presented evidence that *sbb* expression is essential for the survival of a subset of serotonergic neurons, which are lost if *sbb* is lost at the critical period before imaginal metamorphosis. A notable analogy to the neuronal loss due to *sbb* loss observed in this study is the massive cell death reported in the developing wing disc induced by mutations in *optomotor-blind* (*omb*), a gene encoding a transcription factor also acting downstream of Dpp; Omb transcriptionally represses the Dpp receptor gene *thickvein* (*tkv*), and failure in this repression results in *tkv* upregulation, which ultimately causes cell death[22]. Importantly, Omb requires Sbb to repress *tkv* expression in the larval wing disc[22]. We postulate that Sbb similarly functions to shut down Dpp signalling in the serotonergic *dsx* neurons at a defined time window before the imaginal metamorphosis for proper differentiation of these neurons. The BMP family proteins to which Dpp belongs are conserved across phyla, thereby playing central roles in a variety of morphogenetic events, including sexual character development. For example, the formation of papillary processes in the anal fin, a masculine sexual characteristic in Medaka fish, is induced by androgen via the enhancement of BMP7 expression[23]. Thus, there may exist interesting parallels between vertebrate and invertebrate development in that BMP and the sex-determinant machinery coordinate to specify male-specific characteristics with roles in mating.

In contrast to the developmental role of *sbb*[+], 5-HT produced by *Trh*-positive *dsx* neurons in the VNC is required for the control of copulation in the adult stage. The exact role of serotonergic *dsx* neurons in the VNC in the control of copulatory mechanisms remains to be clarified. Higher-resolution motion recordings of a fly attempting copulation might help in detecting subtle yet critical changes in the specific movements of abdominal structures in *sbb* mutant males.

Although in this study we focused on VNC neurons, it has been demonstrated that activities in a specific cluster of *dsx* single-positive neurons called pC2l (ref. 24) in the brain trigger copulation behaviour[5]. Also, there is evidence of the involvement of the brain serotonergic system in regulating motivation to copulate[25,26]. These brain neurons appear to turn ON and OFF,

via as yet unidentified descending neurons, the VNC motor centre for the control of copulation, whose development and function are likely regulated by the *Trh-GAL4*-positive *dsx* neurons we found in this study. Once initiated, copulation seems to be accomplished by 'copulation-executor' VNC neurons, which may also determine the duration of copulation. Some such VNC neuron groups that affect the copulation duration have been identified, that is, *fru*-positive serotonergic motoneurons innervating internal reproductive organs[27], corazonin-expressing interneurons and their associated downstream serotonergic motoneurons[28], and *dsx*-expressing GABAergic interneurons[6]. A factor affecting the copulation duration is the level of 'persistence', which can be assessed by the effectiveness of a disturbance stimulus to curtail ongoing copulation. An interrupting mating test, performed to measure the ease with which males could be separated from females upon 60-s vortexing, previously revealed that a disturbance stimulus applied during the first 5-min period after the copulation initiation is least effective in forcing a pair to dissociate, and thus that the level of persistence is highest over that time period[3]. The persistence level then gradually decreases over the subsequent 10–15 min, and the copulation ultimately terminates without any disturbance[3]. A later work suggested that the copulation persistence is negatively regulated by the eight *dsx*-positive GABAergic interneurons in the VNC and positively regulated by a dopaminergic system[6]. Our manipulation of serotonergic neurons in the VNC, in contrast, had no discernible impact on persistence of copulation, once the flies started copulation. These considerations give rise to the supposition that copulation is controlled by a multilayered circuitry in which *dsx*-expressing and/or aminergic neurons predominate. 5-HT also plays key roles in the central and peripheral systems for control of copulation in humans and other vertebrates[29], and thus highlights interesting evolutionary parallels across the animal kingdom.

## Methods

**Drosophila stocks.** Flies were reared on cornmeal-yeast medium under a 12:12 light:dark cycle at 25 °C, except for those carrying a transgene for RNA interference (RNAi), which were raised at 29 °C. For the heat-shock induction of *sbb* double-stranded RNA (Fig. 1d), larvae or pupae in food vials were exposed to heat shock of 37 °C for 45 min or 1 h at the indicated developmental timing. The following strains were established, or kept as lab stocks in our laboratory: the wild-type *Canton-S*, +; *plt/SM1*; *ry506*, *elavC155-GAL4*, *N630 hs-GAL4*, *w*; *UAS-mCD8::GFP*, *w*; *UAS-GFPNZ* and *w*;; *fruNP21/TM3 Ser. yw*; *FRT42 sbb6/CyO Act-GFP*, *yw*; *sbbk00702/CyO* and *P{GT1}sbbBG01610/CyO* were the kind gifts from Tetsuya Tabata (University of Tokyo)[12]. *fruGAL4*, *fruFLP*, *UAS > stop > mCD8::GFP*, *UAS > stop > TNT*, *UAS > stop > TNTin* and *UAS > stop > dTrpA1* were the kind gift from Barry J. Dickson (HHMI-Janelia Farm Research Campus)[30,31]. *dsxGAL4*, *dsxFLP* and *elav-GAL80* were the kind gifts from Stephen F. Goodwin (University of Oxford)[17,32]. *repo-GAL80* was the kind gift from Leslie B. Vosshall (Rockefeller University)[6]. *fruLexA* was the kind gift from Bruce S. Baker (HHMI-Janelia Farm Research Campus)[33]. *Otd-FLP* was the kind gift from David J. Anderson (California Institute of Technology)[34]. *tsh-GAL80* was the kind gift from Julie Simpson (HHMI-Janelia Farm Research Campus). *w*;; *poxn-GAL4-14-1-7*, *UAS-mCD8::GFP* was a gift from Ken-ichi Kimura (Hokkaido University of Education). *w*, *ppk-GAL4* was the kind gift from Kazuo Emoto (University of Tokyo). *w*;; *GAD-GAL4* on 3rd was the kind gift from Yuh Nung Jan (University of California at San Francisco)[35]. The following strains were obtained from the Bloomington Stock Center: *w1118*; *Df(2R)ED3610/CyO* (#9066), *w1118*; *Df(2R)BSC334/CyO* (#24358), *w1118*; *Df(2R)ED3683/SM6a* (#8918), *w1118*; *Df(2R)BSC335/CyO* (#24359), *w1118*; *Df(2R)Exel7153/CyO* (#7893), *w1118*; *Df(2R)BSC399/CyO* (#24423), *w1118*; *Df(2R)BSC483/CyO* (#24987), *w*; *Tdc2-GAL4* (#9313), *w*;; *TH-GAL4* (#8848), *Vglut-GAL4* on X (#24635), *w1118*; *VglutOK371-GAL4 (II)* (#26160), *w*; *Cha-GAL4*, *UAS-GFP* (#6793), *w*; *Trh-GAL4* (#38388), *w*; *GAD-GAL4* on 2nd (#51630), *w*; *UAS-Dcr2* (#24650), *w*;;*UAS-Dcr2* (#24651), *tubP > GAL80 >*; *Bl/CyO*; *TM2/TM6* (#38879), *w*; *tubP > GAL80 >/CyO*; *TM2/TM6* (#38880), *w*; *Sp/CyO*; *tubP > GAL80 >/TM6B* (#38881), *w*; *tubP > stop > GAL80 >*; *MKRS/TM6* (#38878), *w*; *wgSp-1/CyO*; *tubP > stop > GAL80* (#39213), *w*; *UAS > stop > mCD8::GFP*; *UAS > stop > mCD8::GFP* (#30032), *w*;; *LexAop-GAL80* (#32213) and *UAS-Trh-RNAi* (#25842). The *w1118;UAS-sbbRNAi* (#41845) strain was obtained from the Vienna *Drosophila* Resource Center.

**Behavioural assays.** The virgin males and females were collected at eclosion. Males were placed singly in food vials, while 10 females were placed together in single food vials. They were kept at 25 °C until being subjected to behavioural assays. For the 5-HTP treatment (Fig. 4f), virgin males carrying both *Trh-RNAi* and *dsx-GAL4* were collected upon emergence, and kept singly with food containing 100 mM 5-HTP[20]. Control flies were fed on normal food without 5-HTP throughout the larval and adult stages. The flies were kept at 25 °C until behavioural assays. Behavioural assays were carried out on males aged 5–7 days after eclosion. The male was placed in a small chamber (0.8 cm in diameter, 0.3 cm in height) with a wild-type virgin female (5–7 days after eclosion). The mating behaviour was recorded for 60 min by a charge-coupled device camera (WAT-221S, Watec, Japan). The courtship activity was quantified by the courtship index, which was defined as the proportion of time a male spent for courtship within an observation period (see below). Actions considered to be elements of courtship included: orientation, following, tapping, wing extension, licking and attempted copulation. The observation period started just after the introduction of the virgin female into the observation chamber. The observation period for the courtship index calculation was 10 min, or until the time when copulation started within the 10 min observation period. The mating success was calculated as the number of pairs that copulated during the observation period (60 min) divided by the number of total pairs observed. In every experiment, we chose a sample size similar to that adopted in preceding works so as to ensure adequate power to detect a meaningful difference among the data sets. The experiments were replicated by at least two different researchers. The statistical analyses were carried out by the Kruskal–Wallis test with Steel–Dwass multiple pair-wise *post hoc* comparisons for the courtship index, or by the $\chi^2$-test for the mating success.

**Immunohistochemistry.** The CNS was dissected from flies in PBS, and fixed in 4% paraformaldehyde in PBS for 60 min. Immunostaining was carried out as described previously[16], using the following antibodies at the indicated dilutions: rabbit anti-GABA diluted at 1:500 (Sigma), rabbit anti-5-HT at 1:500 (Sigma), chicken anti-GFP at 1:1,000 (Abcam), rabbit polyclonal anti-GFP at 1:1,000 (Molecular Probes) and mouse monoclonal nc82 at 1:20 (DSHB, University of Iowa, IA). The secondary antibodies used were as follows: Alexa647-conjugated goat anti-mouse IgG at 1:200, Alexa546-conjugated goat anti-rabbit IgG at 1:200, Alexa488-conjugated goat anti-chicken IgG at 1:200, Alexa488-conjugated goat anti-rabbit IgG at 1:200 and Alexa546-conjugated goat anti-mouse IgG at 1:200 (all from Invitrogen). Stacks of optical sections were obtained with a Zeiss LSM 510 META confocal microscope and were processed with Image J software.

**Data availability.** The authors declare that the data supporting the findings of this study are available within the article (and its Supplementary Information files), or available from the authors on request.

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

## Acknowledgements

We are grateful to D. Anderson, B. Baker, B. Dickson, K. Emoto, S. Goodwin, Y. Jan, K. Kimura, J. Simpson, T. Tabata and L. Vosshall for fly stocks. We thank G. Toba for technical guidance in the early stage of this work. This work was supported in part by Grants-in-Aid for Scientific Research (nos 16H06371, 15K14306, 26113702, 26114502 and 23220007) from MEXT to D.Y. and a scholarship offered by MEXT to Y.B.Y. (144019).

## Author contributions

J.X. and D.Y. conceived the project and experimental design. Y.B.Y., M.K. and K.S. performed the experiments. Y.B.Y and D.Y. wrote the paper.

**Additional information**

**Competing financial interests:** The authors declare no competing financial interests.

