## [Peer Review File · Nature Communications]

Reviewers' comments:

Reviewer #1 (Remarks to the Author):

In this manuscript Yilmazer et al report on a *Drosophila* mutant line that courts vigorously but doesn't copulate, called platonic (plt), and suggests that a P-element insertion in the gene scribbler (sbb) accounts for this phenotype. The phenotype also includes a wing vein defect as well. The authors suggested further, that a small group of 8 doublesex (dsx)-expressing serotonergic neurons in the abdominal ganglion that degenerated during early development (at day 4 prior to imaginal metamorphosis) in the absence of sbb expression, were missing in adult flies and that that was the ultimate cause of the copulation defect.

This is an interesting suggestion and describes an interesting separation between courtship behavior, copulation attempts and copulation success. There do, however, appear to be a number of issues that should be addressed.

1. There is a relatively large literature on serotonergic neurons, fruitless and doublesex and male sexual behavior in *Drosophila*. There are reported important contributions to copulatory behavior by FRU-expressing neurons in abdominal ganglia, and by FRU/DSX neurons as well. While several of the other published articles are listed in the references in this manuscript, they are not described in any detail in the text. Hence a description should be included of the variety of roles that serotonin is suggested to serve in male copulatory behavior, how different groups of serotonergic neurons in the abdominal ganglion could serve in this variety of roles, and how the DSX-expressing neurons, might actually fit into the story.
2. The authors suggest that there were no "obvious" structural anomalies in the external genitalia in sbbplt hemizygous mutants. How was this evaluated? Did the authors perform detailed histological examination of the genital apparatus, especially regarding the muscular apparatus. It is not clear what "obvious" means. Apparently courtship attempts were reduced by about 50% (suppl. Fig 4) in the hemizygotes compared to CS. Could this be due to muscular defects of some sort?? Was the muscle structure examined?
3. In the RNAi experiments with different transmitter Gal4 drivers (Fig 3d) the greatest effect was seen with one of the two GAD-Gal4 drivers (on 2nd chromosome). In the description of Fig 3d in the text it reports a "marked" reduction in copulation success with the Trh-driver, yet the decrease was much greater with the GAD-Gal4 driver, and Cha-, TH- and v-glut- drivers all showed small but significant reductions. Why wasn't there additional follow-up on the knockdown of sbb by the GAD driver? Why the focus only on the serotonergic neurons?
4. In the experiments in which the authors knocked down Trh expression with RNAi in dsx-expressing cells, the authors generated a phenotype that resembled the sbbplt phenotype (absence of copulation success). Was immunocytochemistry performed of 5HT expression in the brain, thoracic and abdominal ganglia in the nervous systems of these flies? Were message levels of trh measured? How complete was the knockdown? Did it only knock out 5HT synthesis in the group of 8 dsx-neurons the authors are interested in, or were other, perhaps FRU/DSX neurons, also depleted of 5HT? Did the authors examine 5HT immunostaining after the 5HTP feeding? Did the group of 8 neurons now show staining? Did the authors try 5HTP or 5HT feeding in the sbbplt hemizygote mutant flies? There should be no restoration of copulation success in the mutants if the dsx- neurons are required and are no longer present as the authors suggest in the manuscript.

A few smaller points:

1. Apparently immunostaining of Trh-GAL4 positive dsx-expressing cells in flies that also contained UAS-sbbRNAi revealed that "more than half" of these cells were lost by sbb knockdown. Numbers cited were 3.6 in experimentals vs 6.6 in controls. That appears to be less than, not more than half. (page 11)
2. Did the authors mean "identified" and not "deified" four lines up from the bottom of the page on p8.

Reviewer #2 (Remarks to the Author):

The authors started the research by following up a copulation-deficiency mutant called platonic, and provided substantial evidences supporting that BMP signaling regulates the development of ~8 serotonergic abdominal neurons that control copulation. But there are still some concerns that need to be addressed before publication.

(1) Although the authors seems to favor that platonic mutants are only defective in the late step of courtship---attempted copulation or copulation, the behavior index (copulation success) they used is not sophisticated enough to differentiate copulation from early courtship steps like following or singing. The number of attempted copulation seems a better way to go. In addition, it is important to quantify the overall courtship index (e.g. total courtship duration) for several key genetic manipulations, in order to exclude the possibility that copulation failure is not due to decreased courtship vigor.

(2) The authors find that TNT inactivation of dsx+/Trh+ neurons suppresses copulation. It would be interesting to look at the phenotype of TrpA and/or Chrimson activation: Could activation of those neurons elicit voluntary attempted copulation in male flies and increase copulation success when paired with females?

(3) In Figure 3d, it appears that there are no quantitative difference between Vglut-GAL4 on X, Trh-GAL4 and GAD-GAL4 on 3rd, at least the authors did not statistically compare them. Thus it may not be justified to exclude the possible involvement of SN and TN2 neurons by saying that Vglut-GAL4 driven sbbRNAi only weakly inhibited copulation. Moreover, the expression pattern in Figure 3e-h should be confirmed by GABA, 5HT and Vglut antibody staining. It looks like that a fair number of brain dsx neurons (pC1 and pC2) express GABA (3e) and TN2 neurons are triple positive for GABA, 5HT and Vglut. Is that really the case? If the GAL4 drivers do not faithfully recapitulate endogenous expression of these neurotransmitters, the authors should discuss it in the text.

(4) In Figure 2e, 2f, 2g, 3a, 3b, 3c, the authors only show behavior index for those manipulations. I would suggest use those intersectional drivers to drive GFP expression and show a z-projection picture for each of them, just to validate that the manipulations are correct.

Minor suggestions:

(5) The effects of sbbRNAi should be confirmed by RT-PCR.

(6) It would be interesting to test if overexpression of sbb in dsx neurons or 5HT neurons is able to rescue the platonic mutant phenotype.

Reviewer #3 (Remarks to the Author):

In this well-written paper, the authors report an interesting discovery of the neural mechanism of the copulation behavior in *Drosophila* that may have evolutionarily conserved, broad implications of general interest. The authors isolated and studied a new mutant that displays a unique phenotype, i.e. failure of male flies to copulate in spite of vigorous courtship behavioral display. This study leads to identification of a novel gene, platonic (plt) that controls 5-HT expression in a group of defined neurons in the CNS (thoracic-abdominal ventral ganglia). This gene is also functionally linked to a major gene, scribbler, in the BMP family. Therefore this study defines a role of 5-HT neurons, highlighting these small group (7-8) of identified cells in the abdominal segments

of the ventral ganglion, and fills an important gap in the complex neural circuits that control the characteristic sequence of the courtship behavior repertoire, i.e. the final step, copulation.

The authors employed the powerful Gal4-UAS expression system in a series of ingeniously designed experiments to obtain clear-cut results. The genetic analysis was further enhanced by direct feeding of 5-HTP in cell-ablated flies to demonstrate the 5-HT neuro-humoral/modulatory action in copulation. The methodology and experimental protocols have been described in detail to enable reproduction by other researchers in the field. The experiments have been competently performed and data analysis has been appropriately documented. The results are well presented with well-organized, clearly displayed figures.

However, the presentation of this paper may be further improved to enhance the impact of this finding and a few pieces of comparison data or control experiments may be added to fully support the claims in the conclusion. Since the analyses of 5-HT association with copulation success may be relevant to other species across phyla, establishing a firm conclusion here will facilitate future work from a broad readership of this paper.

Specific comments:

1. The introduction may be expanded to include more background or specific information to make the significance of this study more apparent. For example, how well copulation has been studied in comparison with other sequential components in the courtship behavior? It can be more clearly spelled out the significance of uncovering the involvement of 5-HT neurons in copulation.

2. The authors report that *sbb* males showed "little effect on courtship activities (Fig. 2a, b and Supplementary Fig. 5) (p6 line 4)" but was "unsuccessful in copulation (p6 line 5)". But Figure 2a, b and Suppl Figure 5 do not seem to be relevant to the claim. Please cite the appropriate Figures or previously published results to support the statements.

In order to establish and distinguish the phenotypes of *plt*, *sbb* and other relevant mutants, it will be helpful to introduce and analyze, at least semi-quantitatively, the behavioral components in the courtship repertoire, i.e. orientation, following/chasing, tapping, wing extension, licking, attempted copulation or mounting and successful mounting/copulation (p18 line 4-5, Methods). Although these behaviors were said to be observed, only the number of attempted copulation was described (reduced to half). It will be more informative to report any significant changes among any of these behavioral elements, which maybe potentially relevant to the failure of copulation.

Importantly, It is necessary to clearly define the term "copulation success" in this paper. How is it related to just simple mounting behavior? What are the criteria for successful mounting, and successful copulation, e.g. distinct mechanically unseparable coupling?

3. The authors used a number of genetic manipulations to narrow down the important anatomical loci for *sbb* action in copulation behavior. Although supported by 5HTP feeding, the negative data used to eliminating CNS regions containing the anatomical loci should be supported by confirmation of the actions of the drivers employed. Are they actually working in the intended manner? The authors should either cite published references or show data to support their claims. Some examples:

Fig 3a: *otd-FLP* with *tubP>stop>GAL80* combination

Fig 3c: *tsh-GAL80*

Fig2d: *repo-Gal80*

Fig 2g: *fruLexA* with *lexAop-GAL80* combination

Also, references for some of these stocks (e.g. + *fruFLP*) should be added. The authors cited reference 24 and 25 for *GAL80*, but reference 24 may be an error in manuscript preparation.

Minor comments.

1. The 2nd sentence in the abstract needs to be reconstructed for clarity.
2. p5, line4. Fig. 1c should be Fig. 1d.

Point-by-point replies to the reviewers' comments

Reviewers' comments:

Reviewer #1 (Remarks to the Author):

In this manuscript Yilmazer et al report on a *Drosophila* mutant line that courts vigorously but doesn't copulate, called platonic (plt), and suggests that a P-element insertion in the gene scribbler (sbb) accounts for this phenotype. The phenotype also includes a wing vein defect as well. The authors suggested further, that a small group of 8 doublesex (dsx)-expressing serotonergic neurons in the abdominal ganglion that degenerated during early development (at day 4 prior to imaginal metamorphosis) in the absence of sbb expression, were missing in adult flies and that that was the ultimate cause of the copulation defect.

This is an interesting suggestion and describes an interesting separation between courtship behavior, copulation attempts and copulation success. There do, however, appear to be a number of issues that should be addressed.

Q1. There is a relatively large literature on serotonergic neurons, fruitless and doublesex and male sexual behavior in *Drosophila*. There are reported important contributions to copulatory behavior by FRU-expressing neurons in abdominal ganglia, and by FRU/DSX neurons as well. While several of the other published articles are listed in the references in this manuscript, they are not described in any detail in the text. Hence a description should be included of the variety of roles that serotonin is suggested to serve in male copulatory behavior, how different groups of serotonergic neurons in the abdominal ganglion could serve in this variety of roles, and how the DSX-expressing neurons, might actually fit into the story.

A1. We have extended the Discussion section to describe more about the known roles of different classes of serotonergic neurons and dsx-expressing neurons in mating behavior (line 3, P. 19 – line 14, P. 20).

Q2. The authors suggest that there were no "obvious" structural anomalies in the external genitalia in *sbb* hemizygous mutants. How was this evaluated? Did the authors perform detailed histological examination of the genital apparatus, especially regarding the muscular apparatus. It is not clear what "obvious" means. Apparently courtship attempts were reduced by about 50% (suppl. Fig 4) in the hemizygotes compared to CS. Could this be due to muscular defects of some sort?? Was the muscle structure examined?

A2. We examined the genitalia and associated muscles in two mutant genotypes and a heterozygous control, *plt/sbb*[BG01610], *plt/Df(2R)BSC334*, and *plt/+*, and found that neither the genitalia nor the muscles exhibit discernible anomalies in their structures. These results are now shown in Supplementary Figure 4.

Q3. In the RNAi experiments with different transmitter Gal4 drivers (Fig 3d) the greatest effect was seen with one of the two GAD-Gal4 drivers (on 2nd chromosome). In the description of Fig 3d in the text it reports a "marked" reduction in copulation success with the *Trh*-driver, yet the decrease was much greater with the GAD-Gal4 driver, and *Cha*-, *TH*- and *v-glut*- drivers all showed small but significant reductions. Why wasn't there additional follow-up on the knockdown of *sbb* by the GAD driver? Why the focus only on the serotonergic neurons?

A3. The effect of *sbb* knockdown in GAD-GAL4-expressing cells was equally strong as that in *Trh*-GAL4 expressing cells. However, as stated in the original manuscript, GAD-GAL4 drove *sbb* RNAi expression in hundreds of cells in contrast to *Trh*-GAL4, which drove *sbb* RNAi expression in ~8 cells, implying that there may exist a few serotonergic cells that play key roles in regulating copulation. For this reason, we focused on serotonergic neurons. In addition, because another research group was studying the roles of GABAergic neurons in the regulation of copulation, we thought it would be more appropriate to examine serotonergic neurons. In this revision, we conducted double staining of the CNS with an anti-serotonin antibody and a reporter for GAD-GAL4 and *Vglut*-GAL4 in addition to *Trh*-GAL4, and found that some of the cells positive for GAD-GAL4 or *Vglut*-GAL4 also express serotonin (Supplementary Figure 10). This finding suggests that the observed impairment of copulation by *sbb*

RNAi when driven via GAD-GAL4 or Vglut-GAL4 could in part be due to RNAi effects on these serotonergic neurons with GAD-GAL4/Vglut-GAL4 expression. We have shown CNS images doubly stained with the marker for Trh-GAL4, GAD-GAL4 or Vglut-GAL4 and the anti-serotonin antibody in Supplementary Figure 10.

Q4. In the experiments in which the authors knocked down Trh expression with RNAi in dsx-expressing cells, the authors generated a phenotype that resembled the sbbpl phenotype (absence of copulation success). Was immunocytochemistry performed of 5HT expression in the brain, thoracic and abdominal ganglia in the nervous systems of these flies? Were message levels of trh measured? How complete was the knockdown? Did it only knock out 5HT synthesis in the group of 8 dsx-neurons the authors are interested in, or were other, perhaps FRU/DSX neurons, also depleted of 5HT? Did the authors examine 5HT immunostaining after the 5HTP feeding? Did the group of 8 neurons now show staining? Did the authors try 5HTP or 5HT feeding in the sbbpl hemizygote mutant flies? There should be no restoration of copulation success in the mutants if the dsx- neurons are required and are no longer present as the authors suggest in the manuscript.

A4. In the experiment shown in Figure 4f, we knocked down Trh in all dsx-GAL4-positive cells. In the present revision, we conducted anti-serotonin antibody staining of the VNC in flies expressing dsx-GAL4, UAS-GFPNZ and UAS-Trh RNAi (Supplementary Figure 12). The results showed that the number of 5-HT and GFP double-positive cells is reduced in these flies compared with the control flies and that the number of cells with detectable immunoreactivity increases with 5-HTP feeding in flies with Trh knockdown (Supplementary Figure 12). These results are now described in the text (lines 13-16, P. 15 and lines 12-16, P. 16). We have found, in contrast, that 5-HTP feeding cannot rescue the copulatory defects in flies with sbb knockdown. Please see also A9.

A few smaller points:

Q5. 1. Apparently immunostaining of Trh-GAL4 positive dsx-expressing cells in flies that also contained UAS-sbbRNAi revealed that "more than half" of these cells were

lost by *sbb* knockdown. Numbers cited were 3.6 in experimentals vs 6.6 in controls. That appears to be less than, not more than half. (page 11)

A5. We rephrased this as “nearly half”.

Q6. 2. Did the authors mean "identified" and not "deified" four lines up from the bottom of the page on p8.

A6. We corrected “deified” to “identified”.

Reviewer #2 (Remarks to the Author):

The authors started the research by following up a copulation-deficiency mutant called *platonic*, and provided substantial evidences supporting that BMP signaling regulates the development of ~8 serotonergic abdominal neurons that control copulation. But there are still some concerns that need to be addressed before publication.

Q7. (1) Although the authors seems to favor that *platonic* mutants are only defective in the late step of courtship---attempted copulation or copulation, the behavior index (copulation success) they used is not sophisticated enough to differentiate copulation from early courtship steps like following or singing. The number of attempted copulation seems a better way to go. In addition, it is important to quantify the overall courtship index (e.g. total courtship duration) for several key genetic manipulations, in order to exclude the possibility that copulation failure is not due to decreased courtship vigor.

A7. We have added data for the courtship index obtained in flies with *sbb* knockdown (Supplementary Figure 7). A quantitative analysis of attempted copulation and several other courtship parameters (courtship index, wing extension and licking) was made for different allelic combinations of *sbb* mutants, as is now shown in Supplementary Figure 5. These results indicate that *sbb* mutant males retain high levels of courtship vigor

while showing a reduction in attempted copulation, consistent with the hypothesis that *sbb* is uniquely required for successful copulation.

Q8. (2) The authors find that TNT inactivation of *dsx+/Trh+* neurons suppresses copulation. It would be interesting to look at the phenotype of *TrpA* and/or *Chrimson* activation: Could activation of those neurons elicit voluntary attempted copulation in male flies and increase copulation success when paired with females?

A8. We performed an experiment to activate *dsx+/Trh+* neurons with *dTrpA1*, and found that there was no increase in copulation success when the resulting male flies were paired with females. Solitary males with *dTrpA1*-activated *dsx+/Trh+* neurons did not display copulation behavior. These results seem to suggest that *dsx+/Trh+* neurons function to tune copulation behavior rather than to induce it by providing sufficient amounts of serotonin.

Q9. (3) In Figure 3d, it appears that there are no quantitative difference between *Vglut-GAL4* on X, *Trh-GAL4* and *GAD-GAL4* on 3rd, at least the authors did not statistically compare them. Thus it may not be justified to exclude the possible involvement of SN and TN2 neurons by saying that *Vglut-GAL4* driven *sbbRNAi* only weakly inhibited copulation. Moreover, the expression pattern in Figure 3e-h should be confirmed by GABA, 5HT and *Vglut* antibody staining. It looks like that a fair number of brain *dsx* neurons (*pC1* and *pC2*) express GABA (3e) and TN2 neurons are triple positive for GABA, 5HT and *Vglut*. Is that really the case? If the GAL4 drivers do not faithfully recapitulate endogenous expression of these neurotransmitters, the authors should discuss it in the text.

A9. We have conducted double labeling experiments of the VNC with each GAL4 reporter and the corresponding neurotransmitter. Among the antibodies tested, anti-serotonin and anti-GABA antibodies worked well, whereas antibodies against L-glutamate or *Vglut* did not yield reliable staining in our hands. It turned out that the “neurotransmitter-specific” GAL4s used in this study do not always faithfully recapitulate the localization of relevant neurotransmitters, which was the result anticipated from other GAL4-UAS experiments. In the process of performing these

experiments, we obtained evidence that TN2 is negative for both GABA and 5HT, as summarized in the Results section (line 16, P. 13 – line 4, P. 14) and the associated Supplementary Figures 10 and 11. Our careful examination with the anti-GABA antibody also revealed that neither the population of pC1 nor that of pC2 included GABAergic neurons, although some of these neurons were labeled by GAD-GAL4 (on 2nd) and Vglut-GAL4 (onX). These GAL4 drivers also labeled small subsets of *dsx*-expressing neurons designated pCd and pLN (white arrows in Supplementary Figure 11; nomenclature based on Kimura et al., 2015). We further confirmed that pC1, pC2, pCd and pLN were entirely negative for 5-HT based on staining with the anti-5-HT antibody. Please see also A4.

Q10. (4) In Figure 2e, 2f, 2g, 3a, 3b, 3c, the authors only show behavior index for those manipulations. I would suggest use those intersectional drivers to drive GFP expression and show a z-projection picture for each of them, just to validate that the manipulations are correct.

A10. We performed an experiment to label the cells in which GAL4 is active in flies of the same genotypes as used for region-restricted *sbb*-knockdown. The results are shown in Supplementary Figure 8, and verify our notion that *sbb* knockdown was achieved in a head-excluded, head-specific or thorax-excluded manner, and thus the behavioral effects shown in Figure 2 can be ascribed to *sbb* knockdown in the targeted cells as planned.

Minor suggestions:

Q11. (5) The effects of *sbb*RNAi should be confirmed by RT-PCR.

A11. We confirmed by RT-PCR that RNAi effectively knocked down *sbb* mRNAs, as shown in Supplementary Figure 2.

Q12. (6) It would be interesting to test if overexpression of *sbb* in *dsx* neurons or 5HT neurons is able to rescue the platonic mutant phenotype.

A12. We attempted to rescue the *plt* phenotype by the overexpression of UAS-*sbb*⁺, yet without success. We presume that the appropriate balance in the amounts of the multiple

isoforms and/or exact timing of their expression may be pivotal for rescuing the phenotype. In our previous study, we experienced a similar difficulty in rescuing the fru mutant behavioral phenotypes by UAS-fru+ overexpression.

Reviewer #3 (Remarks to the Author):

In this well-written paper, the authors report an interesting discovery of the neural mechanism of the copulation behavior in *Drosophila* that may have evolutionarily conserved, broad implications of general interest. The authors isolated and studied a new mutant that displays a unique phenotype, i.e. failure of male flies to copulate in spite of vigorous courtship behavioral display. This study leads to identification of a novel gene, platonic (plt) that controls 5-HT expression in a group of defined neurons in the CNS (thoracic-abdominal ventral ganglia). This gene is also functionally linked to a major gene, scribbler, in the BMP family. Therefore this study defines a role of 5-HT neurons, highlighting these small group (7-8) of identified cells in the abdominal segments of the ventral ganglion, and fills an important gap in the complex neural circuits that control the characteristic sequence of the courtship behavior repertoire, i.e. the final step, copulation.

The authors employed the powerful Gal4-UAS expression system in a series of ingeniously designed experiments to obtain clear-cut results. The genetic analysis was further enhanced by direct feeding of 5-HTP in cell-ablated flies to demonstrate the 5-HT neuro-humoral/modulatory action in copulation. The methodology and experimental protocols have been described in detail to enable reproduction by other researchers in the field. The experiments have been competently performed and data analysis has been appropriately documented. The results are well presented with well-organized, clearly displayed figures.

However, the presentation of this paper may be further improved to enhance the impact of this finding and a few pieces of comparison data or control experiments may be added to fully support the claims in the conclusion. Since the analyses of 5-HT association with copulation success may be relevant to other species across phyla,

establishing a firm conclusion here will facilitate future work from a broad readership of this paper.

Specific comments:

Q13. 1. The introduction may be expanded to include more background or specific information to make the significance of this study more apparent. For example, how well copulation has been studied in comparison with other sequential components in the courtship behavior? It can be more clearly spelled out the significance of uncovering the involvement of 5-HT neurons in copulation.

A13. Thank you very much for the encouraging comments and thoughtful suggestions. We have expanded the Introduction section to include more background knowledge on the mechanism of copulation and the importance of serotonin involvement in the regulation of copulation (line 1, P. 3 – line 11, P. 4). We have also expanded the Discussion section to mention the different roles of distinct groups of serotonergic neurons in *Drosophila* mating (line 3, P. 19 – line 14, P. 20). Please see also A1.

Q14. 2. The authors report that *sbb* males showed "little effect on courtship activities (Fig. 2a, b and Supplementary Fig. 5) (p6 line 4)" but was "unsuccessful in copulation (p6 line 5)". But Figure 2a, b and Suppl Figure 5 do not seem to be relevant to the claim. Please cite the appropriate Figures or previously published results to support the statements.

A14. We added the data for the courtship index of flies of the relevant genotypes (Supplementary Figures 6 and 7). The relevant passage was changed to the following: "In contrast to the partial suppression of copulation success with *fru*-GAL4 (Fig. 2a), *dsx*-GAL4 completely blocked copulation when used to express UAS-*sbb*RNAi (Fig. 2b and Supplementary Fig. 6), with little effect on courtship activities (Supplementary Fig. 7a, b)". (lines 5 – 9, P. 8).

Q15. In order to establish and distinguish the phenotypes of *plt*, *sbb* and other relevant mutants, it will be helpful to introduce and analyze, at least semi-quantitatively, the

behavioral components in the courtship repertoire, i.e. orientation, following/chasing, tapping, wing extension, licking, attempted copulation or mounting and successful mounting/copulation (p18 line 4-5, Methods). Although these behaviors were said to be observed, only the number of attempted copulation was described (reduced to half). It will be more informative to report any significant changes among any of these behavioral elements, which maybe potentially relevant to the failure of copulation.

A15. In the revised manuscript, we included quantitative comparisons of total courtship activities (as measured by the courtship index), wing extension and licking in addition to the data for attempted copulation, which were already included in the original submission (Supplementary Figure 5). These results illustrate the overall pattern of courtship activities in *sbb* mutants, which is quantitatively different from that of the wild type.

Q16. Importantly, It is necessary to clearly define the term "copulation success" in this paper. How is it related to just simple mounting behavior? What are the criteria for successful mounting, and successful copulation, e.g. distinct mechanically unseparable coupling?

A16. In our observation, male flies deficient in *sbb* function either failed in mounting or exhibited persistent copulation once they had succeeded in mounting. In no case did male flies deficient in *sbb* function curtail their copulation after successful mounting. We therefore refer to persistent copulation (a copulation duration of ~15-20 min) as successful copulation. Nonetheless, our analysis does not allow us to specify that the affected step in *sbb*-deficient flies is mounting per se, and therefore, we prefer to use the phrase "copulation success" rather than, for example, "successful mounting".

Q17. 3. The authors used a number of genetic manipulations to narrow down the important anatomical loci for *sbb* action in copulation behavior. Although supported by 5HTP feeding, the negative data used to eliminating CNS regions containing the anatomical loci should be supported by confirmation of the actions of the drivers employed. Are they actually working in the intended manner? The authors should either cite published references or show data to support their claims. Some examples:

Fig 3a: otd-FLP with tubP>stop>GAL80 combination

Fig 3c: tsh-GAL80

Fig2d: repo-Gal80

Fig 2g: fruLexA with lexAop-GAL80 combination

A17. We have labeled the GAL4-positive cells in the respective flies with UAS-mCD8::GFP, verifying the notion that sbb knockdown was done in a head-excluded, head-specific, thorax-excluded, neuron-excluded or glia-excluded manner. These results are shown in Supplementary Figure 8. Please see also A10.

Q18. Also, references for some of these stocks (e.g. + fruFLP) should be added. The authors cited reference 24 and 25 for GAL80, but reference 24 may be an error in manuscript preparation.

A18. The relevant references are cited in the revised manuscript.

Minor comments.

Q19. 1. The 2nd sentence in the abstract needs to be reconstructed for clarity.

A19. The relevant portion of the Abstract has been rewritten.

Q20. 2. p5, line4. Fig. 1c should be Fig. 1d.

A20. The error has been corrected.

REVIEWERS' COMMENTS:

Reviewer #1 (Remarks to the Author):

Manuscript # NCOMMS-16-05872A

Title: The *Drosophila* platonic mutant: why doesn't he copulate?

Reviewer 1 comments:

Almost all of the issues raised by this reviewer have been adequately responded to by the authors. I have a few suggestions of wording changes that might improve the manuscript, but they needn't delay acceptance of the revised manuscript:

1. Line 15 (and several other places in the manuscript - see line 246, 274). Use of the word "remarkably" in the sense of uncommon or extraordinary. I see nothing extraordinary about a defect in 5HT leading to a copulation defect, since the authors stress several places in the manuscript that 5HT is involved in other animal species in the regulation of sex specific behaviors. Maybe just leave remarkable or remarkably out---possibly substitute interesting?
2. Line 100 might be a little strong about "no structural anomalies in external genitalia and associated muscles" as shown in Supplementary Figure 4 and its discussion. For example, were muscle fiber numbers counted? was muscle strength evaluated? How complete an examination was made? Perhaps just say "no obvious structural anomalies, etc."
3. Line 227-228 might be made more accurate by changing "likely trigger degeneration of these cells" to "likely triggers degeneration of some of these cells" since it appears that there is a random loss of about half of the cells with reduced *sbb* expression.

Reviewer #3 (Remarks to the Author):

The authors have made substantial efforts to resolve the issues raised in the previous review. The presentation of the revised manuscript has been greatly improved to consolidate and communicate its interesting findings. The revised paper is acceptable for publication.

Point-by-point replies to the reviewers' comments

Reviewer 1 comments:

Almost all of the issues raised by this reviewer have been adequately responded to by the authors. I have a few suggestions of wording changes that might improve the manuscript, but they needn't delay acceptance of the revised manuscript:

Q1. Line 15 (and several other places in the manuscript - see line 246, 274). Use of the word "remarkably" in the sense of uncommon or extraordinary. I see nothing extraordinary about a defect in 5HT leading to a copulation defect, since the authors stress several places in the manuscript that 5HT is involved in other animal species in the regulation of sex specific behaviors. Maybe just leave remarkable or remarkably out--possibly substitute interesting?

A1. We removed remarkably/remarkable in lines 15 and 246, and replaced it with interesting in line 274.

Q2. Line 100 might be a little strong about "no structural anomalies in external genitalia and associated muscles" as shown in Supplementary Figure 4 and its discussion. For example, were muscle fiber numbers counted? was muscle strength evaluated? How complete an examination was made? Perhaps just say "no obvious structural anomalies, etc."

A1. We rephrased this passage as "...no obvious structural anomalies,...".

Q3. Line 227-228 might be made more accurate by changing "likely trigger degeneration of these cells" to "likely triggers degeneration of some of these cells" since it appears that there is a random loss of about half of the cells with reduced sbb expression.

A3. We rephrased this passage as "likely triggers degeneration of some of these cells".

Reviewer #3 (Remarks to the Author):

Q4. The authors have made substantial efforts to resolve the issues raised in the previous review. The presentation of the revised manuscript has been greatly improved to consolidate and communicate its interesting findings. The revised paper is acceptable for publication.

A4. Thank you for your great support.